# Profiling Attached Residents in an Urban Community in the U.S.: An Empirical Study of Social–Landscape Interactions within a Park

**Ying Xu [1,2,3], Jae Ho Lee [4,*] and David Matarrita-Cascante [5]**

1  Department of Tourism Management, Shaanxi Normal University, No. 620, Chang'an Street, Chang'an, Xi'an 710119, China; xuying129@snnu.edu.cn
2  Shaanxi Tourism Information Engineering Laboratory, No. 620, Chang'an Street, Chang'an, Xi'an 710119, China
3  Shaanxi Key Laboratory of Tourism Informatics, No. 620, Chang'an Street, Chang'an, Xi'an 710119, China
4  Department of Landscape Architecture, University of Seoul, Seolusiripdae-ro 163, Dongdaemun-gu, Seoul 02504, Korea
5  Department of Rangeland, Wildlife and Fisheries Management, Texas A&M University, College Station, TX 77843, USA; dmatarrita@tamu.edu
*  Correspondence: jaeho19@uos.ac.kr

**Abstract:** Community attachment has been studied predominantly in terms of the social dimensions of community life, which explains what makes residents feel connected to a locality. Following a more recent trend within the community attachment literature, this study examined the role of communities' physical dimensions in fostering sentiments of rootedness and connections to communities. More specifically, the study sought to better understand the role that urban parks play in predicting community attachment using a discriminant analysis technique to profile attached residents. We selected Discovery Green Park in Houston, Texas, as the study site, where we administered 606 total surveys to park visitors inquiring about their interactions with the park itself, emotional connections to it, and social interactions within the park. We found that strongly attached residents tend to be older, have a greater reliance on the park service and programs, and have meaningful interactions with new people in the park and frequently visit the park in groups to socialize and relax. Additionally, those who are strongly attached to the community attribute greater symbolic meanings to Discovery Green and more strongly identify with the park. The profile of residents attached to the community, given their interactions with the park and its visitors, provides important knowledge to both park managers and community leaders; they can use this information to create conditions, fostering more strongly attached residents who tend to be active agents of positive change in the community.

**Keywords:** community attachment; urban parks; urban community; discriminant analysis; place identity

## 1. Introduction

Community scholars have long been interested in understanding the emotional bonds to specific localities and the implications of such bonds for community life (Brehm 2007; Kasarda and Janowitz 1974). In understanding such bonds, community research has examined the notion of 'community attachment,' which refers to people's emotional connection to their community through feelings of rootedness and belonging (Matarrita-Cascante et al. 2010).

Beyond romanticized notions of attachment, the relevance of people's sentimental bonds with their communities can produce desirable outcomes (Brehm et al. 2006; Cross 2003; Hummon 1992; Matarrita-Cascante 2014). For instance, community attachment is associated with improved quality of life (Goudy and Ryan 1982), decreased depression (O'Brien et al. 1994), increased consumption of locally produced food (Cowell and Green 1994), job satisfaction (Apostle et al. 1985), social development (Hummon 1992), community participation (Gooch 2003; Kelly and Hosking 2008; Kyle et al. 2010; Pradhananga

and Davenport 2017; Theodori 2004, 2018), well-being (Grobecker 2016; Theodori 2001), support for local tourism development (Bajrami et al. 2020; Campón-Cerro et al. 2017; Eslami et al. 2019), and pro-environmental behaviors (Brehm et al. 2006; Safshekan et al. 2020; Soleimani and Nohegar 2019). Matarrita-Cascante (2014) have concluded that the implications of community attachment are not limited to simple emotional outcomes or desires but instead significantly relate to a localities' social, economic, and environmental well-being. Thus, community attachment is an important subject of inquiry for academics and practitioners, given its multifaceted impact on community life.

Traditionally, researchers have examined communities by focusing on the role of social relations (e.g., number and intensity of friendships), concluding that people become attached to their communities due to their interpersonal connections. More recently, researchers have included the role of the physical place in predicting community attachment (Arnberger and Eder 2012; Brehm et al. 2004, 2006; Clark and Stein 2003; Matarrita-Cascante 2014; Matarrita-Cascante et al. 2010; Trentelman 2009). For example, Matarrita-Cascante et al. (2010) found that natural environment-based interactions and residence choice motivations (linked to natural amenities) resulted in increased community attachment for both long-term residents and seasonal residents in natural resource-rich areas. Similarly, Xu et al. (2019) have shown that community attachment is not determined completely by social factors but also by physical environmental factors.

The present study builds on earlier work (e.g., Brehm et al. 2004, 2006; Clark and Stein 2003; Matarrita-Cascante et al. 2010; Xu et al. 2019) that suggests the physical environment's importance in predicting community attachment. Most studies taking this newer approach, however, have focused on the role of the natural environment with respect to amenity-rich rural communities. This study takes the additional step of identifying individuals attached to an urban community (an urban park, specifically) through their interactions with the local physical environment. Thus, this research aims to answer the question 'What is the profile of residents attached to urban communities?' Knowing the characteristics of attached residents can facilitate their identification, which is vital due to their improvement of community well-being.

## 2. Framework for Analysis

Traditional studies of community attachment have focused on social aspects (e.g., social interaction, friendship). Accordingly, scholars have measured community attachment using two main models. First, there is the linear model, which holds that community attachment is determined primarily by structural characteristics of a locality (e.g., size; Kasarda and Janowitz 1974). This model hypothesizes that a community's increasing size, density, and heterogeneity will result in decreased levels of community attachment (Goudy 1990; Gusfield 1978; Kasarda and Janowitz 1974; Theodori and Luloff 2000). However, scholars have become increasingly critical of this model from a theoretical perspective, and it has consistently failed empirical tests in numerous studies (Goudy 1990; Gusfield 1978; Theodori and Luloff 2000).

Second, the systemic model focuses on the predictive values of three systemic factors (length of residence, social class, and age) and two groups of intervening variables (amity and associational bonds; Kasarda and Janowitz 1974; Matarrita-Cascante et al. 2010). Length of residence's role in increasing emotional connections to the community has emerged in many studies, given its ability to foster associational bonds and amity, specifically through localized social interactions with friends, family, and other community members. That is, community attachment is linked strongly to the social ties people develop in the community, which tend to strengthen with time (Beggs et al. 1996; Goudy 1990; Kasarda and Janowitz 1974).

More recently, studies have noted the role of non-social factors (e.g., the physical environment) in predicting community attachment. Brehm (2007) found that several dimensions of the natural environment help define people's community attachment. Matarrita-Cascante et al. (2010) have reported that newcomers to amenity-rich areas rapidly

develop strong sentimental ties to a community based on their experience with the local natural landscape. The human-natural landscapes connectedness, including various dimensions of experiential, cognitive and emotional (Riechers et al. 2020), have been recognized as closely related to people's sense toward local communities (Li et al. 2021; Peng 2020). More specifically, Chang et al. (2020) has identified the effects of urban green areas in improving the consciousness and support of local communities by encouraging their residents to conduct outdoor recreational activities. Furthermore, as suggested by the previous literature, urban green space contributed to enhancing social interactions and cohesion among residents, thereby furthering their sense of community (Arnberger and Eder 2012; Cooper et al. 2014; Dipeolu et al. 2020; Talen 2000).

In response to the findings supporting the significance of the physical or natural attributes in fostering community attachment, scholars have established a connection to the place literature (Bow and Buys 2003; Corcoran 2002; Theodori and Luloff 2000; Trentelman 2009). Matarrita-Cascante et al. (2010, pp. 213–14) have noted:

> *'On a more theoretical level, our research has utilized traditional community attachment measures, and has expanded this framework somewhat by including natural resource-related motivations and interactions. Further expansion, however, is in order. Sense of place research uses the same core construct (place attachment), but (a) expands this construct to potentially include multiple domains (e.g., identity and dependence); (b) broadens the potential mechanisms by which attachment is built to explicitly include interactions with the natural environment; and (c) analyzes the contribution of descriptive place meanings to attachment. This literature has its own nexus of variables and operational measures that are related to, yet remain somewhat distinct from, those in the community-based literature. We did not make use of the sense of place literature in creating our measures, but believe this area of inquiry holds great potential for combining with community to form more synthetic understandings . . . '*

As noted above, the place attachment literature traditionally has focused on individuals' connections with specific areas without considering the cumulative effects that such connections may produce in the larger social context—local communities. Conversely, the community literature typically has examined the social bonds developed in a local area without considering place-based values. Despite such discrepancies between community and place literature (Trentelman 2009), scholars have pointed out the potential of place theory to enhance our understanding of individuals' complex combination of feelings towards local communities. According to Hummon (1992), while community attachment is most strongly associated with localized social networks, it is shaped to some extent by individuals' subjective perceptions of locally constructed environments. McKnight et al. (2017) has also argued that individuals inevitably imbue social meanings to the physical place where they live and interact with others. This helps them establish a sense of place, which in turn affects their community experience and community sentiments.

Given the physical landscape's potentially great contribution to community attachment, some recent research (e.g., Arnberger and Eder 2012; Brehm 2007; Brehm et al. 2006; Matarrita-Cascante et al. 2010) has incorporated this variable within the measurement of community attachment. Community attachment is not predicted solely by social factors; it also depends on residents' emotional connections and interactions with the local physical environment (Xu et al. 2019).

In most sociologic research, researchers examining community attachment have emphasized the importance of social ties. Hence, our current state of knowledge is still limited regarding the physical/natural environment's contribution to individuals' attachment to their communities. In an effort to expand our knowledge of this topic, this research aims to explore the contributions of urban physical/natural landscape to community attachment. To refine the approaches previous research adopted to measure natural environment-oriented attachment (e.g., Brehm et al. 2004; Matarrita-Cascante et al. 2010), this study expands the framework by targeting multiple landscape-related factors: respondents' interactions with the landscape through the engagement of recreational activities, respondents'

spiritual and emotional bonds to the natural landscape, and social interactions occurred within landscape settings.

In addition, scholars have argued that communities vary in the extent to which community attachment is related to certain physical and social features (Goudy 1990; Sampson 1988). The predominant focus of these studies has been natural amenity-rich rural areas. Whether incorporating physical/natural related factors in community attachment measures are applicable to urban contexts. In response, by selecting parks as a typical form of physical/natural landscape in cities, this study extends earlier work by identifying individuals attached to an urban community based on their relationships with an urban park (with both natural and built attributes).

Furthermore, our study sheds light on the link between the place and the community literature by utilizing the notion of a sense of place as a measure of emotional connections with the physical/natural landscape to explore if and how much it predicts ones' community attachment levels, which further recognizes the importance of local physical/natural landscape to community life. This study investigates the characteristics of attached individuals based on different forms of human interactions and relationships associated with parks. This is particularly relevant considering the rapid, global trend of urbanization (Ritchie and Roser 2018) and attached residents' role in fostering community well-being.

## 3. Methods

### 3.1. Sample Selection and Data Collection

To profile attached residents based on their interactions with urban physical space, we collected data by administering a survey instrument between 30 June and 18 July 2015, at Discovery Green Park in Houston, Texas. Located downtown, Discovery Green is a well-known and active urban park that attracts numerous visitors year-round ('Discovery Green', n.d.), as it provides a variety of leisure activities and serves as a space for social gatherings. Such opportunities result from the park's features, including its unique landscape, which showcases the value of green space in urban Houston.

To secure a representative sample of participants at Discovery Green, we collected data on both weekdays and weekends from 11:00 a.m. to 6:00 p.m., which included peak- and off-hours (provided by Discovery Green Conservancy). We recruited samples in various locations within the park: Gateway Fountain, McGovern Playground, Brown Foundation Promenade, Kinder Lake, The Lake House (restaurant), and Sarofim Picnic Lawn. During each sampling period, two researchers approached park visitors at random and informed them of the research objective. Following Babbie's (2009) ethical guideline, all participation was voluntary, and we guaranteed strict confidentiality. Those who agreed to participate received a paper questionnaire to complete in the field. If participants did not want to complete the questionnaire in the field, we sent a link to an online version of the survey via email two days after having approached them in the park. The online version of the survey was identical to the one administered in the park. We sent a reminder email with a link to the questionnaire seven days after the initial posting in July 2015. Additionally, we printed signs including a link to the survey and posted them in various locations within the park as well as on Discovery Green's Twitter and Facebook pages to recruit more participants.

During the field data collection period, the researchers contacted 733 total individuals in person and via email, 712 of whom agreed to participate (face-to-face survey n = 600, online survey n = 112). Among the 600 individuals who agreed to participate and received printed copies, 546 returned the questionnaires in the park, and 20 questionnaires were incomplete, yielding 526 complete responses. For the online surveys, 112 participants opened the link posted on social media or signs in the park, and 80 completed the survey. The response rates for each method were 71.8% and 71.4%, respectively. Ultimately, we analyzed 606 total responses.

### 3.2. Measurement Approach

3.2.1. Dependent Variable: Community Attachment

We measured respondents' perceived level of attachment to their community through a battery of Likert-scaled items—ranging from 1 ('strongly disagree') to 5 ('strongly agree')—used by Matarrita-Cascante et al. (2010). Respondents indicated their levels of agreement with five statements regarding community attachment (Table 1). We conducted exploratory factor analysis (EFA) using direct oblimin extraction (Table 1), and the results indicated the existence of a single dimension comprised of the five items. The total variance explained was 68.01%, and the Cronbach's alpha of the underlying dimension was 0.88.

**Table 1.** Factor loadings and reliability for items measuring community attachment.

| Construct | M | SD | Factor Loading | Eigenvalue | % Variance |
|---|---|---|---|---|---|
| **Community attachment (Cronbach's $\alpha$ = 0.88)** | | | | 3.40 | 68.01 |
| I feel this community is a real home to me | 3.69 | 0.95 | 0.71 | | |
| I feel I belong to this community | 3.72 | 0.91 | 0.79 | | |
| I feel I am fully accepted as a member of this community | 3.70 | 0.91 | 0.78 | | |
| Most people in this community would go out of their way to help me if I was in trouble | 3.36 | 0.91 | 0.60 | | |
| Most of the people in this community can be trusted | 3.27 | 0.93 | 0.52 | | |

Extraction method: direct oblimin.

The resulting measure yielded average values that ranged from 0 to 5, where higher values stronger attachment to greater Houston. Because the normal distribution was slightly skewed toward strong attachment (mean = 3.55), we created a dichotomous variable indicating high (3.55 to 5) versus low (0 to 3.54) levels of attachment.

3.2.2. Independent Variables

- Place-based sociodemographic variables

Place-based variables are usually used in studies of large-scale urban communities where socioeconomic and other demographic disparities exist (Elnakat et al. 2016; Fortney et al. 2000). Such studies have commonly utilized ZIP code data, which show differences in neighborhood characteristics within the broader city. Therefore, to account for the statistical impact of disparities in Houston's living areas, we requested ZIP codes from survey respondents to identify their neighborhoods' characteristics. We then utilized secondary data sources to develop a dataset of place-based variables consisting of housing characteristics, public transportation, and distance to the park.

First, we accessed 2013–2017 data from the American Community Survey (ACS) at the ZIP code level (ZIP code tabulation area, or ZCTA) to identify housing characteristics in greater Houston. We first discovered that single-family homes accounted for the largest proportion of Houston's homes (57.90%). The variable was then expressed as a percentage of the total for each ZIP code to describe Houston's housing characteristics. Second, we collected several shapefiles for the most popular transportation modes to Discovery Green (i.e., rail stations, bus routes, and bike routes) via City of Houston GIS (COHGIS) and overlapped them on a GIS map. We overlaid them on a map containing Houston's city limits to account for the possible effect of accessibility to transportation to the park on community attachment. Then, we indexed these modes from 0 (no access to the park) to 3 (access to all) to assign weights to park accessibility. Lastly, we used distance to the park to capture the possible impact of residential locations on community attachment (Nelson et al. 2006). To aggregate the dataset at the ZIP code level by distance to the park, we stratified distance to the park based on geographical proximity (1 = 0–4.99 miles; 2 = 5–9.99 miles; 3 = 10–14.99 miles; 4= more than 15 miles). We used these three indicators as place-based variables in our study.

- Socio-demographics

Our questionnaire collected sociodemographic data from respondents, including age, gender, highest education level completed, current employment status, and household annual income (see Table 2). We coded respondents' ages in 10-year increments to make comparisons between age groups. We measured gender dichotomously (0 = male, 1 = female). We measured education level with six ordinal categories: 1 = less than high school degree, 2 = high school degree or GED, 3 = some college, 4 = trade/technical/vocational training or associate's degree, 5 = four-year college/ Bachelor's degree, and 6 = advanced degree (Master's, Ph.D., JD, MD). We measured race/ethnicity with six categories: 1 = White or Anglo, 2 = Hispanic or Latino, 3 = Black or African American, 4 = Native American or American Indian, 5 = Asian or Pacific Islander, 6 = other. We measured employment status dichotomously (1 = employed, 0 = unemployed). Finally, we measured income with 10 categories: 1 = less than $10,000, 2 = $10,000 to $14,999, 3 = $15,000 to $24,999, 4 = $25,000 to $34,999, 5 = $35,000 to $49,999, 6 = $50,000 to $74,999, 7 = $75,000 to $99,999, 8 = $100,000 to $149,999, 9 = $150,000 to $199,999, and 10 = $200,000 or more.

**Table 2.** A sociodemographic overview of the sample.

| Variables | N | % |
|---|---|---|
| **Gender** | | |
| Male | 168 | 28.1 |
| Female | 429 | 71.9 |
| **Age** | | |
| 19–29 | 187 | 31.1 |
| 30–39 | 214 | 35.5 |
| 40–49 | 110 | 18.3 |
| 50–59 | 38 | 6.3 |
| 60–69 | 20 | 3.3 |
| Over 70 | 5 | 0.8 |
| **Education** | | |
| Less than a high school degree | 27 | 4.5 |
| High school degree or GEDs | 116 | 19.3 |
| Some college | 128 | 21.3 |
| Trade/technical/vocational training or associate degree | 74 | 12.3 |
| 4-year College/University Bachelor's degree | 147 | 24.4 |
| Advanced degree (Master's, Ph.D., JD, MD) | 106 | 17.6 |
| **Employment status** | | |
| Employed for wages | 355 | 55.6 |
| Self-employed | 73 | 12.1 |
| Out of work and looking for work | 30 | 5.0 |
| Out of work but not currently looking for work | 14 | 2.3 |
| A homemaker | 70 | 11.6 |
| A student | 45 | 7.5 |
| Military | 1 | 0.2 |
| Retired | 11 | 1.8 |
| Unable to work | 3 | 0.5 |
| Other | 10 | 1.7 |
| **Household annual income** | | |
| Less than $10,000 | 44 | 7.3 |
| $10,000 to $14,999 | 26 | 4.3 |
| $15,000 to $24,999 | 31 | 5.1 |
| $25,000 to $34,999 | 42 | 7.0 |
| $35,000 to $49,999 | 91 | 15.1 |
| $50,000 to $74,999 | 125 | 20.8 |
| $75,000 to $99,999 | 57 | 9.5 |
| $100,000 to $149,999 | 74 | 12.3 |
| $150,000 to $199,999 | 46 | 7.6 |
| $200,000 or more | 32 | 5.3 |

- Interactions with the physical landscape in the urban park

Respondents' interactions with natural areas have been commonly measured as engagement in outdoor recreation activities (Matarrita-Cascante et al. 2010; Moore and Graefe

1994; Ryan 2005; Williams and Roggenbuck 1989). Building on previous studies, we measured respondents' interactions with Discovery Green in terms of 13 types of recreational activities (Table 3). Respondents indicated the extent to which they had participated in these activities at Discovery Green Park during the last 12 months (1 = never, 2 = rarely, 3 = sometimes, 4 = often, 5 = always). We instructed first-time visitors to skip this question. We conducted principal components analysis (PCA) to condense the 13 items into similar conceptual categories. We deleted four items (4, 5, 8, 9) because of our cutoff value of 0.4 (Matsunaga 2010). Thus, the PCA procedure yielded three categories of recreational activities, which we labeled as (a) passive activities (four items), (b) park-sponsored activities (three items), and (c) children-oriented activities (two items). The total variance explained was 68.54%, and the Cronbach's alphas of the three underlying dimensions were 0.80, 0.76, and 0.69, respectively.

**Table 3.** Factor loadings and reliability for items measuring independent variables.

| Construct | M | SD | Factor Loading | Eigenvalue | % Variance |
|---|---|---|---|---|---|
| **Interactions with physical landscape** | | | | 6.17 | 68.54 |
| Passive activities (Cronbach's α = 0.80) | | | | 3.82 | 42.42 |
| Socializing with family and/or friends | 2.84 | 1.31 | 0.64 | | |
| Visiting gardens | 2.09 | 1.23 | 0.64 | | |
| Walking/jogging/running | 2.09 | 1.34 | 0.66 | | |
| No specific activity, just enjoy a nice day out in the park | 3.01 | 1.45 | 0.67 | | |
| Park-sponsored activities (Cronbach's α = 0.76) | | | | 1.30 | 14.47 |
| Attending concerts/movies/shows | 1.78 | 1.08 | 0.77 | | |
| Special events/festivals | 2.12 | 1.17 | 0.72 | | |
| Fitness classes | 1.33 | 0.80 | 0.50 | | |
| Children-oriented activities (Cronbach's α = 0.69) | | | | 1.05 | 11.65 |
| Playing around the fountain area | 2.70 | 1.47 | 0.78 | | |
| Children's programming and/or play | 2.10 | 1.38 | 0.79 | | |
| **Emotional connections with the urban park** | | | | | |
| Place meanings (Cronbach's α = 0.90) | | | | 5.21 | 52.22 |
| A place to escape the pressure of urban life | 4.02 | 0.93 | 0.69 | | |
| A place to appreciate the beauty of nature | 4.09 | 0.91 | 0.76 | | |
| A place to participate in outdoor recreational activities | 4.23 | 0.81 | 0.72 | | |
| A place for citizens' well-being | 4.12 | 0.82 | 0.76 | | |
| A place to meet friends and socialize | 4.10 | 0.87 | 0.70 | | |
| A place that develops positive feelings about the community | 4.29 | 0.80 | 0.78 | | |
| A place representing the image of Houston | 4.10 | 0.92 | 0.74 | | |
| A place for tourists to visit | 4.26 | 0.86 | 0.72 | | |
| A window into the diversity of traditions of Houston | 4.00 | 0.97 | 0.73 | | |
| A fun place for children to play | 4.62 | 0.70 | 0.61 | | |
| Place identity (Cronbach's α = 0.91) | | | | 3.12 | 78.09 |
| This park means a lot to me | 3.69 | 0.93 | 0.88 | | |
| I am very attached to this park | 3.39 | 0.98 | 0.91 | | |
| I strongly identify with this park | 3.45 | 0.94 | 0.91 | | |

**Table 3.** *Cont.*

| Construct | M | SD | Factor Loading | Eigenvalue | % Variance |
|---|---|---|---|---|---|
| I have special connections to this park and the people who visit the park | 3.15 | 0.97 | 0.84 | | |
| Place dependence (Cronbach's α = 0.92) | | | | 3.23 | 80.72 |
| I enjoy visiting this park more than any other park | 3.63 | 1.01 | 0.82 | | |
| I get more satisfaction out of visiting this park than from any other park | 3.60 | 1.03 | 0.87 | | |
| Visiting this park is more important than visiting any other park | 3.32 | 1.01 | 0.85 | | |
| I would not substitute other parks for the activities I do here | 3.38 | 1.00 | 0.70 | | |
| **Social interactions within the urban park** | | | | | |
| Group size | | | | | |
| Average group size during park visits | 2.90 | 0.89 | | | |
| Interactions with family and friends (Cronbach's α = 0.85) | | | | 2.30 | 76.62 |
| Family and friends in my household | 3.77 | 1.91 | 0.82 | | |
| Family members outside my household | 3.16 | 2.00 | 0.92 | | |
| Friends outside my household | 3.11 | 1.94 | 0.89 | | |
| Interactions with new people at the park (Cronbach's α = 0.78) | | | | 2.40 | 60.02 |
| Chatting with other dog owners when walking my dog | 1.87 | 1.24 | 0.70 | | |
| Playing informal sports or games with others | 1.93 | 1.13 | 0.76 | | |
| Talking with other people while attending concerts, dance parties, or other special events | 2.43 | 1.30 | 0.77 | | |
| Talking with other people while exercising, participating in fitness classes, or other programming offered in the park | 2.01 | 1.26 | 0.86 | | |

Extraction method: direct oblimin.

- Emotional connections with the urban park

    We measured respondents' emotional connections with the park's landscape in terms of symbolic meanings that visitors attributed to the park (i.e., place meaning) and affective bonds between people and the environment (i.e., place attachment), as have been measured in multiple studies (Stedman 2002, 2003; Kudryavtsev et al. 2012).

    Drawing on relevant literature (Stedman 2002), we measured place meaning using a 10-item scale (Table 3; 1 = strongly disagree, 2 = disagree, 3 = neither disagree nor agree, 4 = agree, 5 = strongly agree). Factor loadings exceeded 0.61 (Table 3), and the total variance explained was 52.12%. The scale's Cronbach's alpha was 0.90, indicating high internal consistency.

    For the variable place attachment, we utilized an eight-item scale (1 = strongly disagree, 2 = disagree, 3 = neither disagree or agree, 4 = agree, 5 = strongly agree) following Kyle et al. (2005) (see Table 3). Table 3). As with Kyle et al. (2004, 2005), the structure of place attachment produced two dimensions (place identity and place dependence). The factor loadings exceeded 0.75 for place identity and 0.78 for place dependence (see Table 3). Place identity explained 78.09% of the total variance, and place dependence explained 80.72%. The dimensions had Cronbach's alphas of 0.91 and 0.92, respectively, indicating high internal consistency in both cases.

- Social interactions within the urban park

    We adapted the measures of social interactions within the urban park from previously validated research; they have been used commonly in the literature (Sirina et al. 2017). Following Peters et al. (2010), respondents indicated the average group size during their

park visits (1 = alone, 2 = 1–2 other people, 3 = 3–5 other people, 4 = 6–10 other people, 5 = more than 10 other people). Respondents also indicated how often they socially interacted in the park with each of three different groups (1 = never, 8 = several times a week). We conducted EFA with these three items, and a single dimension emerged, which we labeled as interactions with family and friends. Factor loadings exceeded 0.82, the total variance explained was 76.62%, and the Cronbach's alpha of 0.85 indicated high internal consistency. Additionally, respondents indicated how often they interacted with new people in the park in each of six different scenarios (1 = never to 5 = always; Table 3). We instructed respondents who had never talked to new people in the park to skip this question. We then conducted EFA with these six items. We eliminated two items, yielding a single dimension that we labeled as meaningful interactions with unknown people. Factor loadings exceeded 0.70, the total variance explained was 60.02%, and Cronbach's alpha was 0.78.

*3.3. Data Analysis*

In this study, we profiled attached residents based on their interaction with the physical environment measured along five dimensions: (1) place-based factors (housing type, transportation, distance to the park); (2) socio-demographics (age, employment, income); (3) interaction with the physical environment (passive activities, park-sponsored activities, children-oriented activities); (4) emotional connections with the urban park (place meaning, place identity, place dependence); and (5) social interactions with other people (group size, interactions with family and friends, meaningful interactions with unknown people). We used discriminant analysis to interpret group differences and classify cases based on the characteristics defining a strongly attached resident, addressing missing items with regression imputation was used for missing items. Thus, the analysis classified park visitors as strongly attached and weakly attached residents. We then applied the discriminant function to both groups to ascertain the characteristics defining them. IBM SPSS 26.0 was used to run discriminant analysis.

## 4. Findings

Discriminant analysis showed that individuals with a stronger attachment to the community more strongly identify with the Discovery Green Park environment. Strongly attached residents tend to be older, have a greater reliance on the park service and programs, and are engaged in more children-oriented activities than those who are less attached. Additionally, strongly attached residents exhibited less involvement in park-sponsored activities, resided mostly in non-single family housing units, and attributed symbolic meanings to the park to a greater extent. Furthermore, they had meaningful interactions with new people in the park and frequently visited in groups for socializing and relaxing (Table 4).

The canonical discriminant function, which tested how well the discriminant model fit the data overall (Klecka 1980), indicated that the discriminant function explained nearly 100% of the variation in community attachment (canonical correlation = 0.48; eigenvalue = 0.30). The measure of discrimination of groups using the discriminating variables also reached statistical significance (Wilk's Lambda = 0.76; $\chi^2$ = 150; df = 15; sig = 0.000).

Table 4 shows the contribution of each independent variable to this function. Place identity had the greatest discriminating ability for the strongly attached community members, followed by place dependence, participation in children-oriented activities, and age.

**Table 4.** Canonical Discriminant Function Coefficients (standard coefficients).

| Variable | Discriminant Function |
|---|---|
| Housing type | −0.181 |
| Transportation | −0.003 |
| Distance to the park | −0.040 |
| Age | 0.217 |
| Employment | −0.076 |
| Income | 0.026 |
| Passive activities | 0.009 |
| Park-sponsored activities | −0.189 |
| Children-oriented activities | 0.221 |
| Place meaning | 0.173 |
| Place identity | 0.636 |
| Place dependence | 0.271 |
| Group size | 0.147 |
| Interactions with family and friends | −0.008 |
| Meaningful interactions with unknown people | 0.167 |

The discriminant function correctly classified 68.8% of less attached residents and 73.8% of strongly attached residents (Table 5). The results suggest that this set of characteristics was more informative for identifying strongly attached compared to less attached residents. Additionally, of the 581 cases, 415 cases were correctly classified (192 weakly attached + 223 strongly attached), and 166 cases were incorrectly classified (79 weakly attached + 87 strongly attached). Thus, the discriminant function increased classification accuracy by 71.4% compared with the marginal values for strongly attached and less attached residents.

**Table 5.** Classification Results for Residents (*N* = 581).

| Community Attachment | | Predicted Group Membership | | Total |
|---|---|---|---|---|
| | | Weakly Attached | Strongly Attached | |
| Count | Weakly attached | 192 | 87 | 279 |
| | Strongly attached | 79 | 223 | 302 |
| Percent | Weakly attached | 68.8 | 31.2 | 100.0 |
| | Strongly attached | 26.2 | 73.8 | 100.0 |

## 5. Discussion and Conclusions

With the current rapid urbanization, promoting urban residents' attachment to their localities has received increasing attention from both academics and city planners (Cheng et al. 2021; Knox and Pinch 2000; Theodori 2000; Zhu et al. 2017). This is particularly relevant when understanding the relationship that exists between community attachment and overall community well-being (Forjaz et al. 2011; Nelson and Prilleltensky 2005; Park et al. 2017; Prilleltensky 2005; Theodori 2001).

Community scholars have typically examined community attachment by focusing almost exclusively on social relations, largely ignoring the natural and built environments. Responding to this shortcoming, several recent studies have expanded the definition of community attachment to incorporate the physical landscape (e.g., Brehm 2007; Brehm et al. 2004, 2006; Clark and Stein 2003; Matarrita-Cascante et al. 2010). These studies have identified the natural environment's role in predicting individuals' levels of community attachment, especially in rural areas.

To refine previous research approaches (Xu et al. 2019), this study sought to establish a profile of attached residents by incorporating multiple environment-related factors in an urban setting: respondents' interactions with the park environment through engagement in recreational activities, their emotional bonds to the park's physical environment, and the social interactions occurring within the park.

Through discriminant analysis, we found that place identity had the greatest discriminating ability. Strongly attached residents also strongly identified with the park. This finding aligns with previous theoretical arguments suggesting that promoting place identity associated with the physical environment can increase community attachment (Matsuoka and Kaplan 2008). In the compact and dense urban area of Houston, people's appreciation of physical and natural attributes within Discovery Green evoked feelings toward the park. This manifested in residents' identification with and feelings of belonging to their local communities.

The following important factor in discriminating residents with strong and weak community attachment was place dependence. Place dependence, originally considered 'the perceived strength of association between a person and specific places' (Stokols and Shumaker 1981, p. 457), points to the functional value of a place in satisfying humans' needs given a range of activities provided by the natural place and how it compares with alternative places (Farnum et al. 2005; Halpenny 2006; Jorgensen and Stedman 2001; Stokols and Shumaker 1981; Vaske and Kobrin 2001). This means that compared to other sites, individuals depend more on the social or natural resources of a certain place for desired recreational activities. In our study, respondents were more reliant on the resources, amenities, and unique experiences provided by Discovery Green than those offered by other recreational settings. Place dependence manifested as people's intensive use of the park for certain recreational activities, as well as the park's ability to provide for such use. While in most cases, residents were not aware of their direct park experience in fulfilling certain needs, or they compared it with that found in other communities, such dependence on the particular natural environment would strengthen their emotional bond with the local community, which is known as community attachment.

In the recreation and tourism literature, place identity and place dependence often have been conceptualized and measured as two subdomains of place attachment, which refers to a bond between people and their environment, including positive emotion or affection (Altman and Low 1992; Hummon 1992; Moore and Graefe 1994; Williams et al. 1992). As both place identity (referring to a place that reflects one's personal identity) and place dependence (referring to a place that meets one's goals) significantly discriminated urban residents in terms of their community attachment levels, this study demonstrates that attachment to a community is closely tied to one's emotions toward a particular place in that community. Our findings support the argument that people develop relationships with public places along with attachment to their communities (Dines et al. 2006). In this case, the emotional values evoked by an urban park led individuals to develop an attachment to their local community. This suggests that the emotional components of people–place relationships are relevant to the understanding of sentiments toward the community.

In addition to respondents' emotional connections with the park, one dimension of the construct of individuals' interactions with the park's landscape (participation in children-oriented activities) was significant in differentiating strongly attached residents from weakly attached ones. Parks are one major type of urban green space that is critical for improving the quality of urban life (Chiesura 2004). Much empirical evidence shows that urban parks are capable of satisfying humans' 'nature needs including contact with nature, aesthetic preferences, and recreation and play' (Matsuoka and Kaplan 2008, pp. 9–11). Our study measured individual park landscape interactions by probing respondents' participation in a set of recreational activities at Discovery Green during the previous 12 months. Among those, engagement in children-oriented activities (playing around the fountain area and children's programming and/or play) was important in motivating respondents' visits to the park. This again stresses the utilitarian importance of this park in providing settings for preferred activities. Through such activities, individuals developed 'functional attachment' to the park and compared the quality of life in their community to that in other communities. This was strongly related to a sense of community and attachment. Accordingly, the behavioral indicator of individual-park interactions loaded on the discriminant function describing the characteristics associated with attached urban residents.

Finally, one sociodemographic variable (age) was significant in discriminating residents based on levels of community attachment. As reported previously (e.g., Erickson et al. 2012; Riger and Lavrakas 1981; Stinner et al. 1990; Theodori and Luloff 2000; Trentelman 2009; Ulrich-Schad et al. 2013), age is an important predictor of community attachment, as attachment increases with age. Compared to younger ones, senior residents generally exhibit lower migration rates, being more likely to stay in their communities (Erickson et al. 2012; Ulrich-Schad et al. 2013). The longer individuals stay in a community, the more opportunities they have to develop social connections with other community members and the more interactions they have with the local natural landscape (through recreational activities supported by the natural attributes), which leads to stronger feelings of belonging and rootedness.

This research is not without limitations. First, this study merely investigated local park visitors in an U.S. urban park—Discovery Green Park. This may result in a bias in the research findings since our respondents may be homogenous in terms of their cultural backgrounds and how they experience and feel about the park (Bazrafshan et al. 2021). It would be essential for future research to conduct cross-cultural studies by targeting diverse population groups from different countries or nations and to identify the variability in their attachment levels associated with physical/natural landscapes.

Second, the study was limited to one study site—Discovery Green Park. No comparisons could be made to determine whether the predicting variables examined in this study would present consistent or different powers in other urban parks. Parks have different characteristics in terms of size (Giles-Corti et al. 2005), the number of features and amenities (Kaczynski et al. 2008), presence of sports fields (Floyd et al. 2008), trails (Kaczynski et al. 2008; Reed et al. 2008; Shores and West 2008), and drinking fountains (McCormack et al. 2010), and accessibility (Kaczynski and Henderson 2007). All these attributes are associated with the use of parks and engagement of physical activity at parks (McCormack et al. 2010) for different groups of people. This suggests that parks may elicit from people varying levels of preference and connections. It would be more fruitful for future research to include additional study sites to explore whether these landscape-based factors display consistent predictive values on community attachment across parks.

Third, this study identified the inner-Houston City as a community. Several place-based factors (i.e., housing characteristics, public transportation, and distance to the park) by ZIP code were created, which, however, may indicate an imperfect overview of community characteristics due to the geographic disparities within a large-scale urban community. Thus, further studies were desired to replicate the examination of physical/natural landscapes' contributions to community life in smaller communities.

In conclusion, this study explored the physical environment's role in community attachment in an urban setting. Its aim was to establish a profile of attached residents based on physical environment-related factors. We found common ground with earlier theoretical discussions and studies with respect to the potential contributions of the physical environment in improving community attachment. The discriminant analysis results demonstrated that strongly attached urban residents more strongly identified with the local park environment, exhibited greater reliance on the park and programs, demonstrated more engagement in particular recreational activities (e.g., children's programming and/or play), and were older than less attached individuals. An understanding of these characteristics can help to identify individuals who are interested in promoting and sustaining local environmental qualities, as well as those who can contribute to successful overall community development.

**Author Contributions:** Conceptualization, Y.X.; methodology, Y.X. and D.M.-C.; data curation, J.H.L.; formal analysis, J.H.L.; investigation, Y.X. and J.H.L.; supervision, D.M.-C.; visualization, J.H.L.; writing—original draft, Y.X.; writing—review and editing, D.M.-C. and J.H.L. All authors have read and agreed to the published version of the manuscript.

**Funding:** The open access publishing fees for this article have been supported by The National Natural Science Foundation Project, China, the 'Study on Cultural Space Reconstruction in Tourist Cities from the Perspective of Everyday Heritage—A Case Study of Xi'an' (Grant No. 42001160).

**Institutional Review Board Statement:** The study was conducted according to the guidelines of the Declaration of Helsinki, and approved by the Institutional Review Board of Texas A&M University (IRB2015-0241 and approved on 21 April 2015).

**Informed Consent Statement:** Informed consent was obtained from all subjects involved in the study.

**Data Availability Statement:** Data not available due to ethical restrictions. Due to the nature of this research, participants of this study did not agree for their data to be shared publicly, so supporting data is not available.

**Conflicts of Interest:** The authors declare no conflict of interest.

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
