# Peer review of "Profiling Attached Residents in an Urban Community in the U.S.: An Empirical Study of Social–Landscape Interactions within a Park"

_socsci, doi:10.3390/socsci11010005_

Round 1
Reviewer 1 Report
This research examining the role of urban parks in strengthening community attachment was conducted at Discovery Green Park. This approach, in which physical environmental factors are evaluated in terms of community commitment, has been found to be up-to-date.
The research has been found original in this respect and it is thought to contribute to the literature. However A one-paragraph review article can be added that will emphasize the original aspect of the research and reveal what it differs from similar studies on this subject.
Research questions are valid.
The sample size is sufficient.
The research method and research design are suitable for answering research questions.
The statistical programs and methods used are sufficient.
Reviewer 2 Report
Dear Authors, this is a nice contribution to internationally-important problem. Indeed, it has potential, but additional work is required to bring it in order. I hope following my advice summarized below will make your manuscript looking stronger.
- Title: please, state that you deal with urban parks in the USA.
- Introduction and Section 2: what about literature sources published in the 2020s? I have seen a lot of useful literature published after 2019.
- Subsection 3.1: 2015 – not too old data? I guess that no, but you need to explain this.
- Where is a table summarizing the demographic characteristics of your respondents?
- Which software you have used for calculations?
- Section 5: I think that aesthetics can be considered more extensively in this paper.
- Section 5: what are the limitations of your study? To me, these are two. First, your study reflects only American experience – things may be very different in the other countries like China or Russia. Second, are your findings influenced by any specific characteristics of this particular park and its importance to the local community – would things differ in the case of worse-organized or less-community integrated or urban-periphery parks?
Round 2
Reviewer 2 Report
I'm very satisfied with these revisions.